# JIP2 haploinsufficiency contributes to neurodevelopmental abnormalities in human pluripotent stem cell–derived neural progenitors and cortical neurons

Reinhard Roessler[1] , Johanna Goldmann[1], Chikdu Shivalila[1], Rudolf Jaenisch[1,2]

**Phelan–McDermid syndrome (also known as 22q13.3 deletion syndrome) is a syndromic form of autism spectrum disorder and currently thought to be caused by heterozygous loss of *SHANK3*. However, patients most frequently present with large chromosomal deletions affecting several additional genes. We used human pluripotent stem cell technology and genome editing to further dissect molecular and cellular mechanisms. We found that loss of *JIP2* (*MAPK8IP2*) may contribute to a distinct neurodevelopmental phenotype in neural progenitor cells (NPCs) affecting neuronal maturation. This is most likely due to a simultaneous down-regulation of c-Jun N-terminal kinase (JNK) proteins, leading to impaired generation of mature neurons. Furthermore, semaphorin signaling appears to be impaired in patient NPCs and neurons. Pharmacological activation of neuropilin receptor 1 (NRP1) rescued impaired semaphorin pathway activity and JNK expression in patient neurons. Our results suggest a novel disease-specific mechanism involving the JIP/JNK complex and identify NRP1 as a potential new therapeutic target.**

## Introduction

22q13 deletion syndrome (22qDS) is commonly caused by deletions in the q-terminal end of chromosome 22. Characteristic symptoms of 22qDS include developmental delay, intellectual disability, and severe delay or complete absence of speech (Phelan, 2008). 22qDS is currently seen as synaptopathy, predominantly caused by heterozygous deletion of the gene *SHANK3*. The importance of SHANK3 protein for functional synapses has been robustly established (Durand et al, 2007; Peça et al, 2011; Jiang & Ehlers, 2013; Uchino & Waga, 2013; Mei et al, 2016). Very recently, SHANK3 has been functionally implicated in subcellular signaling pathways, such as the mammalian target of rapamycin (mTOR) pathway, and it has

been shown that SHANK3 depletion impairs intracellular signaling (Shcheglovitov et al, 2013; Bidinosti et al, 2016).

However, the contribution of additional, concomitantly deleted genes in this disease is largely unknown. The vast majority of patients nonetheless present with highly variable chromosomal deletions that can affect several genes located up- or downstream of *SHANK3* (Bonaglia et al, 2011; Sarasua et al, 2014). One interesting candidate gene commonly co-deleted in 22qDS is *MAPK8IP2* (also known as JIP2). JIP2 is known to be a crucial scaffolding protein that facilitates the activities of MAP kinase pathway proteins including mitogen-activated protein kinase kinases, extracellular signaling-regulated kinases, and c-Jun N-terminal kinases (JNKs) (Whitmarsh, 2006). Regulated activation of the MAPK pathway, and in particular correct regulation of JNK proteins, has been shown to be crucial for neuronal differentiation and neuron function (Tiwari et al, 2011; Coffey, 2014a). Interestingly, Tiwari et al (2011) showed that JNK inhibition results in impaired neuronal differentiation of mouse embryonic stem cells (ESCs). It is, therefore, conceivable that JIP2 haploinsufficiency also impacts early human neurodevelopment. Supporting this notion, imaging studies of a small group of Phelan–McDermid patients revealed a neurodevelopmental phenotype predominantly affecting the formation of the cerebellum (Aldinger et al, 2013). Crucially, such a morphological phenotype has not been observed in *Shank3* knockout mice.

Given the complexity of the deletions, we hypothesize that although a small number of patients lacking only *SHANK3* have been diagnosed, 22qDS may frequently not only be the consequence of heterozygous loss of *SHANK3* but might also be linked to the loss of JIP2. In this study, we present preliminary evidence suggesting that both SHANK3 and JIP2 may contribute to disease-specific phenotypes that occur at distinct neurodevelopmental stages. Based on these observations, we propose a potential disease mechanism that involves the MAPK pathway and the regulatory function of JNK proteins and impaired mTOR pathway activity. We believe that the realization of this complex neurodevelopmental and functional

[1]Whitehead Institute for Biomedical Research, Cambridge, MA, USA [2]Department of Biology, Massachusetts Institute of Technology, Cambridge, MA, USA

Correspondence: jaenisch@wi.mit.edu

phenotype is crucial for successful development of potential pharmacological interventions.

# Results

## Patient-specific induced pluripotent stem cells (iPSCs) show impaired neuronal maturation

Patient-specific iPSCs and ESCs were differentiated into neural progenitor cells (NPCs) and mature cortical neurons, and immunoflourescence and immunoblot analysis was used to characterize basal identities (Fig 1A–C). Morphological and molecular profiling revealed that both patient and control pluripotent stem cells (PSCs) could efficiently generate cortical neurons. Both cell populations sequentially differentiated into cells expressing NPC markers such as PAX6 and NESTIN, and continued in vitro maturation resulted in neurons expressing pan-neuronal markers such as TUJ1 and MAP2.

Comparing patient and control neurons, we observed the expected reduction of protein levels for SHANK3 and JIP2 as well as a reduction of neuronal markers such as DCX and NeuN in patient neurons (Fig 1C and D). JNK proteins were only weakly induced in patient neurons as compared with control neurons. Gradual induction of JNK during differentiation of mouse ESCs into neurons has been documented during neural induction (Tiwari et al, 2011). Strikingly, impaired up-regulation of JIP2 in patient neurons coincided with failed induction of JNK proteins (Fig 1C and D). This observation prompted us to investigate electrophysiological properties of mutant cells as an additional criterion for neural maturation. We performed multi-electrode array (MEA) analysis (Maestro; Axion) comparing control neurons and 22qDS neurons during in vitro maturation for up to 95 d (Fig 1H for representative image). Comparative quantification of the neuron content (assessed by the number of MAP2-positive cells) in both conditions did not show any significant deviation (Fig 1E). To ensure equal cell density of neurons plated, we counted the total number of cells (~120,000 cells/well) and evaluated the neuron content by immunocytochemistry of parallel (Fig 1F and G) cultures and Western blot analysis (data not shown). Average spike quantification over a 3-mo period revealed a significant impairment of in vitro maturation in patient-derived neurons (Fig 1I). Although control neurons gradually increased their spontaneous activity, only small surges were observed in patient neurons. Comparative raster plots across 64 individual electrodes at day 71 of differentiation clearly showed higher activity for control neurons (Fig 1J). Quantification of active electrodes across differentiation experiments comparing control and patient neurons also revealed reduced overall activity in 22q13 neurons at the later stages (Fig 1K). By analyzing the mean spike rate during in vitro maturation, we found a gradual increase over time for control neurons (Fig 1L). Together with increasing numbers of active electrodes, this observation indicates an increase in network activity in the control cultures. During the differentiation, time-course spike rates remained low for patient neurons with significant reduction, particularly in late stage mutant neurons. Thus, our results indicate impaired network activity in patient neurons.

Postsynaptic density proteins establish tightly packed aggregates to facilitate efficient synaptic transmission. Furthermore, these structures are directly linked to intracellular downstream signaling pathways (Fig S1A). Protein analysis in 3-mo-old neurons confirmed a severe reduction of SHANK3 in the 22qDS mutant cells associated with reduced levels of direct and indirect binding partners of SHANK3, whereas pan-neuronal markers appear to be unchanged (Fig S1B). In addition, we observed reduced protein levels of protein kinase B (AKT) and mTOR and their phospho-variants in patient neurons. The AKT/mTOR pathway has recently been associated with Phelan–McDermid syndrome (Bidinosti et al, 2016). We also detected reduced levels of Homer and phosphoinositide 3-kinase enhancer (PIKE), which mediate signal transduction to phosphoinositide 3-kinase and eventually trigger the activation of the mTOR complex (Fig S1C). This result confirms that loss of SHANK3 not only leads to impaired synaptic transmission but also influences central intracellular pathways in mature neurons.

## Mutant iPSC-derived neural progenitors reveal a neurodevelopmental phenotype

Our initial observation that 22qDS neurons show severely impaired in vitro maturation led us to explore whether neurodevelopmental defects might be detectable already earlier during the differentiation process and consequently result in functional differences. For this, we genetically engineered fluorescent NPC reporter cell lines for the patient and control background that would allow purification of a defined cell population. We used CRISPR/Cas9 gene targeting to integrate a GFP reporter cassette into the endogenous locus of *PAX6*, a well characterized NPC marker (Osumi et al, 2008; Georgala et al, 2011) (see Fig 2A for targeting strategy). Correctly targeted clones were identified by puromycin selection and Southern blot assays (Fig 2B) and differentiation of these clones into NPCs activated the PAX6-GFP reporter in cells with typical NPC morphology (see representative immunostaining in Fig 2C). Clones with CRISPR/Cas9–induced double-strand breaks in the second allele (PAX6$^{GFP/-}$) resulted in largely GFP-negative cell populations upon differentiation and were, therefore, excluded from further analyses (Fig S2A and B). Separation of GFP-positive and GFP-negative populations by FACS (Figs 2D and S2B) enabled the analysis of global gene expression in a homogenous NPC population. Expression profiling identified a large set of differentially expressed genes comparing the mixed population with the GFP-positive and GFP-negative fraction (Figs 2E and S2C). More specifically, NPC markers were up-regulated in the GFP-positive fraction (Fig 2F), whereas genes up-regulated in the GFP-negative fraction were identified as neural crest markers (data not shown). Gene ontology (GO) analysis revealed the up-regulated population of genes in the NPC fraction as genes characteristic for forebrain progenitors (strongly enriched GO terms are telencephalon and forebrain development, axon guidance, neuron differentiation, etc.) (Fig 2G). This result confirms PAX6 as a robust NPC marker that can be used to purify a defined progenitor population. More detailed transcript analysis showed that NPC markers such as NESTIN, MSI1, ASCL1, and several SOX family members were equally expressed in purified control and patient NPCs (Fig S2D and E).

Figure 1. **Patient-specific iPSCs show impaired neuronal maturation.**
**(A)** Immunostaining for PAX6 and NESTIN in PSC-derived NPCs. **(B)** Immunostaining of PSC-derived cortical neurons for MAP2 and CTIP2. **(C)** Comparative protein analysis for SHANK3 and JIP2 as well as for JNK proteins, NPC- and pan-neuronal markers for control and patient lines. **(D)** Quantification of protein levels in neurons. Data are shown as mean ± SEM (n = 3). Two-tailed unpaired *t* test, *$P < 0.05$. **(E)** Quantification of normalized neuron content plated on MEAs. Data are shown as mean ± SEM (n = 10/11 images per condition, respectively). **(F, G)** MAP2 immunostaining of age/culture-matched (not density-matched) neurons used for MEAs. **(H)** Representative phase-contrast image showing neurons seeded on 64 electrodes/well. **(I)** Quantification of average spike count during in vitro maturation. Average of three wells (three independent cultures) for 5-min measurement intervals are shown as box plots ± SEM. **(J)** Comparative raster plots as recorded on day 71 of differentiation. Every line in an individual plot represents one electrode. Every vertical dash represents a detected field potential. Recording

Having identified specific NPC signatures in the control and patient PAX6-GFP sorted cells, we used this system to compare the expression profiles between purified control and patient NPCs to identify potentially disease-relevant differences. RNA sequencing (RNAseq) analysis showed that relative expression levels of genes within the deletion were largely reduced in patient NPCs and fluctuated around a 0.5-fold expression, whereas expression of a set of NPC markers showed a high level of similarity (Figs 2H and S2F). Global gene expression analysis identified a large set of genes (~300) that were differentially expressed in purified patient NPCs (Fig 2I) and GO analysis classified the down-regulated fraction as genes with biological functions such as "regulation of neurogenesis," "regulation of nervous system development," and "neuron differentiation" (Fig 2J). This result suggests that phenotypic changes in 22qDS NPCs already occur before mature synapses have been established and underlines the notion that mutations in SHANK3 may not be the sole factor determining the pathology in most Phelan–McDermid syndrome patients.

### Isogenic ESC–derived neural cell types recapitulate patient-specific phenotypes

Comparing different iPSC lines to evaluate disease-relevant phenotypes can be particularly challenging as heterogeneous genetic backgrounds may drastically mask subtle cellular phenotypes. Recent advances in genome editing, however, have enabled the generation of isogenic mutant lines that are genetically identical except for the disease relevant deletion or mutation (Soldner et al, 2011; Li et al, 2013). To generate isogenic ESC lines that carry a similar deletion as the patient-specific iPSC line used above, we expressed Cas9 protein and deletion-flanking gRNAs in the WIBR#3 hESC line (see Fig 3A for targeting scheme). We obtained clones that carried a heterozygous deletion of ~93 kb of the long arm of chromosome 22. PCR analysis across or flanking the deletion site identified three heterozygous clones that recapitulated a patient-specific deletion (Fig 3B). Sequence analysis of the wild-type allele showed that the 5′ targeting site contained only a single base pair deletion upstream of the JIP2 gene (Fig 3C). This frame shift mutation, however, did not impact heterozygous expression of JIP2 (Fig 3F and G). Sequencing of the 3′ targeting site on the other hand indicated several insertions/deletions in the isogenic deletion lines (Fig 3C). We found deletion-specific reduction of SHANK3, particularly affecting individual isoforms in isogenic Δ22 lines (Fig 3F and G). Differentiation into NPCs appeared to be normal as assessed by immunostaining for NESTIN and PAX6 (Fig 3D and E). Subsequent differentiation into cortical neurons resulted in equal expression of the pan-neuronal marker TUJ1 in all compared lines (WIBR#3, WIBR#3_Δ22-1, WIBR#3_Δ22-2, and 22q13 iPSCs). Expression of JIP2 and the JIP-binding partner JNK 1/3 was, however, reduced in all deletion lines (Fig 3F and G). This result shows that genetically engineered 22qDS lines reliably recapitulate molecular phenotypes as observed in patient-specific cell types. In addition, we observed reduced levels of DCX in isogenic ESC–derived neurons (Fig 3F and G), confirming the neurodevelopmental molecular phenotype described earlier in iPSC-derived neurons.

### Pharmacological activation of the neuropilin receptor 1 (NRP1) recues JNK expression in maturing patient neurons

Molecular and electrophysiological phenotypes have been described before in the 22qDS mutant cells. However, the contribution of JIP2 haploinsufficiency to 22qDS has not been investigated in human neurons. Maturation of cortical pyramidal neurons depends on robustly established semaphorin signaling, which subsequently activates JNK proteins (Calderon de Anda et al, 2012). Moreover, it has been established that JIP and JNK proteins form functional complexes to facilitate kinase activities (Mooney & Whitmarsh, 2004; Kukekov et al, 2006; Koushika, 2008). We therefore hypothesized that JIP2 haploinsufficiency and consequently reduced JNK activity might contribute to impaired neuronal maturation. To elucidate potentially underlying molecular pathways, we characterized known signaling mediators within the semaphorine pathway. Semaphorins are secreted guidance cues that bind to neuropilin receptors such as NRP1 (Tran et al, 2009; Maden et al, 2012; Pasterkamp, 2012). NRP1 activates the downstream kinase TAOK2 (thousand-and-one amino acid 2 kinase), which in turn phosphorylates and activates JNK proteins, and tightly regulated JNK activity is known to be crucial for, for example, neuronal maturation and axon guidance processes (see Fig 4A for semaphorin pathway scheme) (Koushika, 2008; Coffey, 2014b). RNAseq analysis revealed that, although JIP2 is reduced in patient-derived NPCs, expression levels of TOAK2 and JNK1 are similar when comparing control and patient NPC transcript levels. Strikingly, we observed an increase in NRP1 transcript in 22qDS NPCs (Fig 4B). Except for DCX, only minor transcript reductions were found in a set of direct JNK target genes (Fig 4B). However, protein analysis in neurons showed a decrease of JIP2, TAOK2, JNK1/3, and cJun (Fig 4C and D). These observations are consistent with the possibility that the function of JNK proteins relies on JIP binding, particularly in differentiating neurons. In support of this notion, we find that pharmacological stimulation of the NRP1 receptor by recombinant SEMA3A (i) reverses up-regulation of NRP1 transcripts in patient neurons and restores control levels after 2 h of in vitro exposure and (ii) rescues the expression levels of TOAK2 and JNK but does not stimulate JIP2 expression (Fig 4E). Although these preliminary observations present an interesting potential addition to the molecular underpinnings of 22qDS, definitive causal connections between JIP2 haploinsufficiency and impaired neurodevelopment will have to be addressed in further studies.

## Discussion

We used PSCs to study molecular and cellular phenotypes in NPCs and cortical neurons of 22qDS patients. Admittedly, in part, our study compares ESCs and iPSCs, which by no means constitute optimal disease versus control pairs. However, comparing genetically unmatched iPSCs (e.g., multiple control lines versus multiple patient lines) and their progeny is likely to result in equal discrepancies from line to line as has been proposed in, for example, Choi et al (2015). In future studies, extended analyses of

---

interval: up to 450 s. **(K)** Quantification of number of active electrodes at day 75 and day 85. Data are represented as mean ± SEM. Two-tailed unpaired t test, **P < 0.005. **(L)** Quantification of average spike rate in Hz at early and late stage differentiation. Data are represented as mean ± SEM. Two-tailed unpaired t test, *P < 0.05, **P < 0.005.

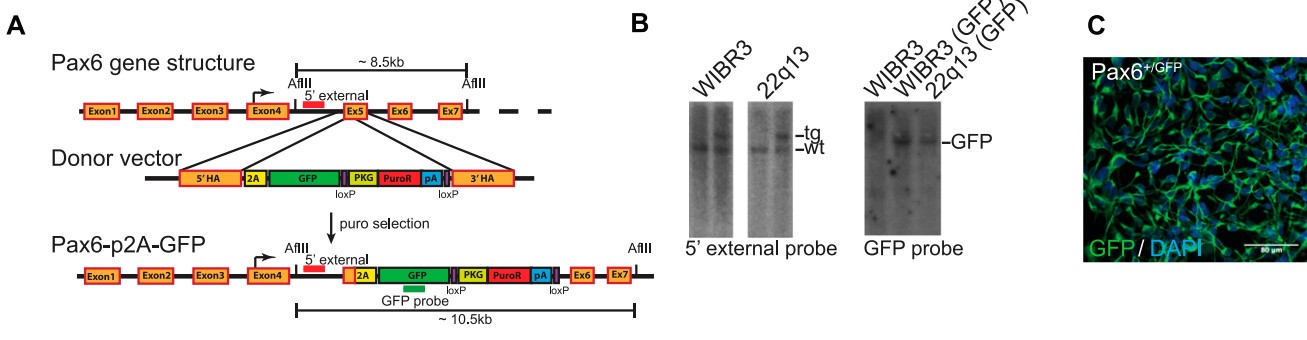

Purification and characterization of Pax6-GFP NPCs

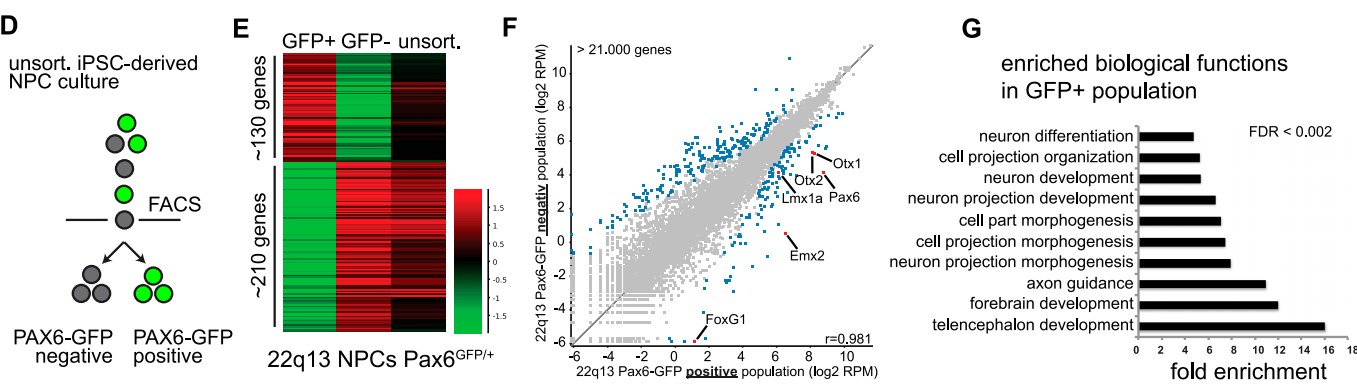

Comparative transcript analysis of patient vs. control NPCs

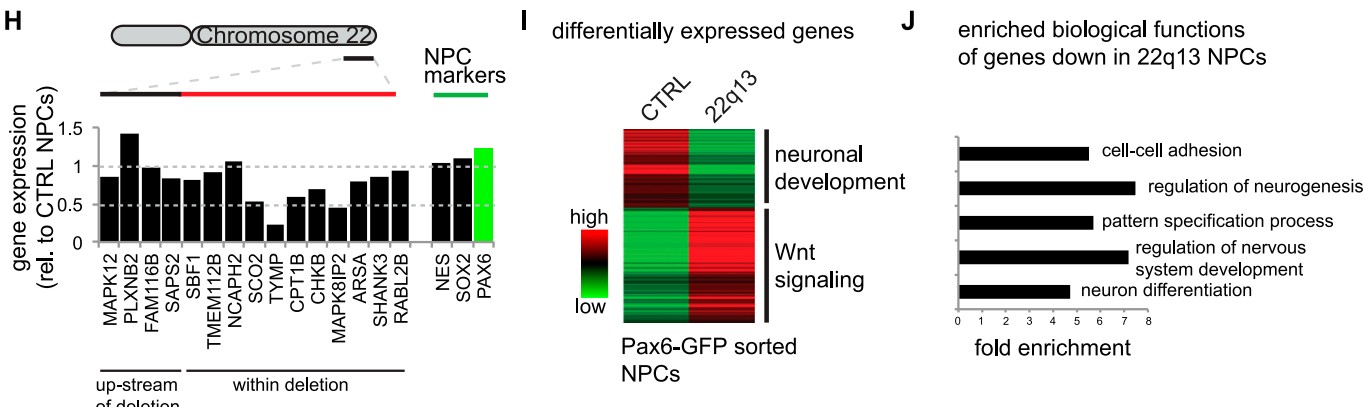

**Figure 2.  Patient neural progenitors show a neurodevelopmental phenotype.**
**(A)** CRISPR/Cas9 targeting scheme for fluorescent PAX6-GFP reporter. **(B)** Southern blot analysis for targeted genomic locus showing 5′ external and internal probe for control and patient specific iPSC lines. **(C)** Representative immunostaining for GFP upon NPC differentiation. **(D)** FACS scheme separating three distinct cell populations. **(E)** Heat map of differentially expressed genes comparing GFP-positive, GFP-negative, and unsorted cell populations. **(F)** Scatter plot representing global gene expression levels. Blue dots represent up-regulated genes in GFP+ and GFP– fraction, respectively. Red dots highlight selected NPC marker genes in the GFP+ fraction. **(G)** GO term analysis of genes up-regulated in GFP+ fraction. **(H)** Quantitation of relative gene expression (rel. to CTRL NPCs) across patient-specific deletion. Selected NPC marker gene expression is shown as reference for PAX6-GFP sorted populations. **(I)** Differentially expressed genes comparing GFP+ cells form controls and patient line. **(J)** GO term analysis of genes down-regulated in patient NPCs (GFP+ cells).

additional patient iPSC lines and isogenic controls (knockout and genetic rescue) might strengthen the findings presented in this study.

Detailed analyses of defined neurodevelopmental stages suggested a novel disease-relevant mechanism that manifests in NPCs and negatively impacts neuronal maturation. Loss of JIP2 may

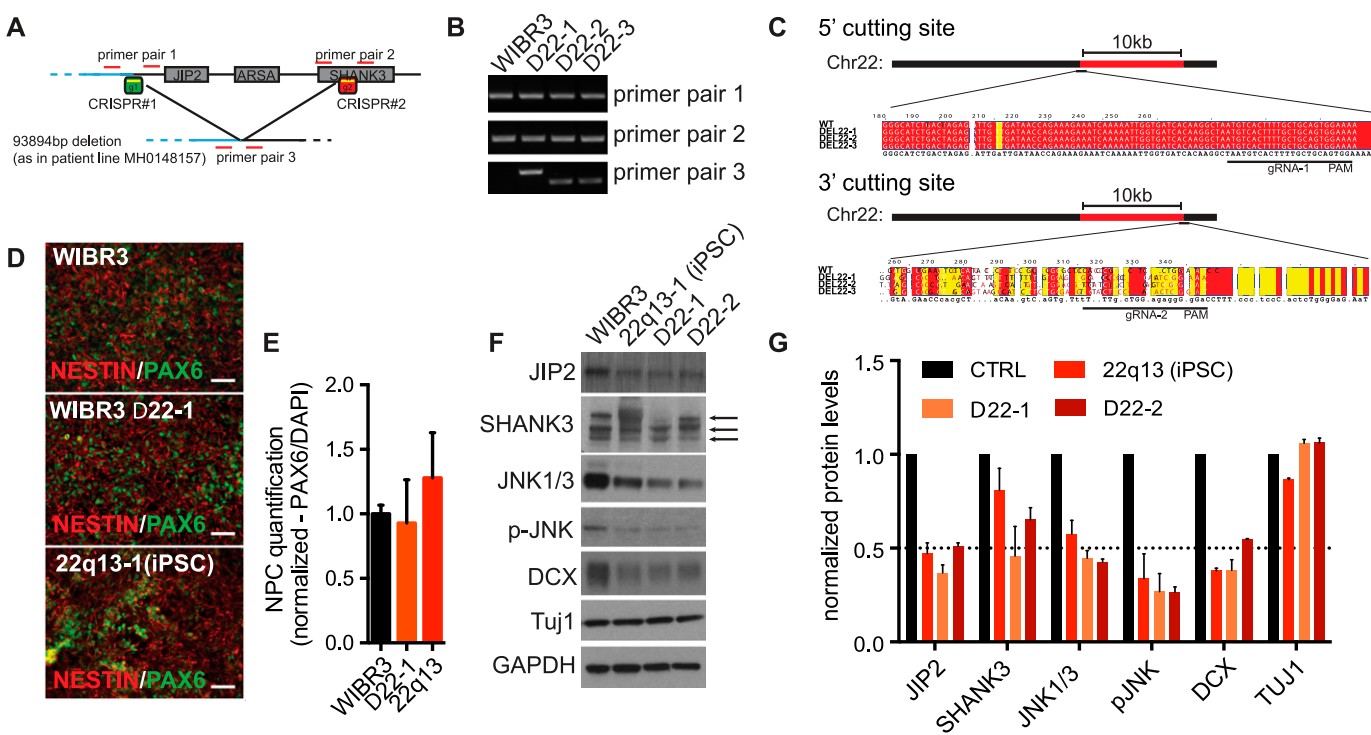

**Figure 3. Isogenic ESC–derived neural cell types recapitulate patient-specific phenotypes.**
**(A)** CRISPR/Cas9 targeting scheme to introduce patient-specific deletion in WIBR#3 ESC line. **(B)** PCR assay shows untargeted control line and three heterozygous deletion lines. **(C)** Genomic DNA sequence of CRISPR target sites flanking the deletion. **(D)** Immunostaining for control, isogenic deletion, and iPSC lines after NPC differentiation. **(E)** Quantification of NPC fraction. Normalized data are shown as mean ± SEM (n = 4). **(F)** Comparative protein analysis of control, isogenic deletion, and iPSC lines upon neuronal differentiation. Arrows indicate three isoforms of SHANK3. **(G)** Quantification of immunoblots shown in (F). Data are represented as mean ± SEM (n = 3).

impact expression levels of a large set of neurodevelopmental genes. Mechanistically, this can be linked to impaired JNK complex function, and pharmacological stimulation of the semaphorin pathway rescues JNK expression. Our study provides indications that JIP2 might contribute to this complex neurodevelopmental disease beyond SHANK3 haploinsufficiency.

22qDS has recently been connected to impaired synaptic transmission and disrupted intracellular signaling (Shcheglovitov et al, 2013; Bidinosti et al, 2016; Lu et al, 2016; Yi et al, 2016). Both cellular phenotypes place SHANK3 at the center of the underlying molecular mechanisms. Although these findings significantly improve our understanding of this complex disorder, it is crucial to note that the vast majority of patients have large chromosomal deletions, which encompass many adjacent genes, consistent with mechanistic causes likely being more complex than currently recognized.

Human PSCs provide a powerful tool to study such complex disorders, particularly if pathogenic genetic mutations have been identified (Soldner & Jaenisch, 2012; Takahashi & Yamanaka, 2013; Soldner et al, 2016). Here, we combined human PSC technology and genome editing techniques to further study and dissect disease-relevant mechanisms. We confirmed findings that identified SHANK3 as a crucial contributor to symptoms associated to 22qDS. Loss of SHANK3 impaired intracellular signaling, in particular the AKT/mTOR pathway, and led to reduced synaptic activity in neurons in vitro. Recently, MEA recordings from Shank3 knockout mouse cortical neurons showed a reduction in network activity (Lu et al, 2016). Our findings in human cortical neurons confirm this observation.

However, impairments seen in patient neurons appear to be more pronounced. The more severe cellular phenotype seen in patient-specific neurons maybe be because of contributions of additional genes that are most commonly deleted in 22qDS. In future experiments, it will be important to validate the MEA results of our current study by single-cell patch-clamp recordings. This should allow consolidation of electrophysiological phenotypes suggested here.

We found that deletion of JIP2 (also known as *MAPK8IP2 or IB2*) likely contributes to an early neurodevelopmental phenotype that can already be detected in neural progenitors. Although the role of JIP2 in autism spectrum disorder (ASD) type diseases is poorly understood, genetic mouse models support a relevant role of this MAPK pathway–related scaffold protein (Giza et al, 2010). Moreover, JIP proteins have been identified as direct binding partners of JNK proteins, which are known to be involved in neuronal maturation and proper neuron function (Mooney & Whitmarsh, 2004; Whitmarsh, 2006; Koushika, 2008; Coffey, 2014b). We found that loss of JIP2 coincided with reduced JNK expression in NPCs and mature neurons, and a detailed analysis of purified iPSC-derived NPCs identified a large set of neurodevelopmental genes to be misregulated. Previous studies described the pivotal role of JNK proteins during in vitro differentiation of mouse ESCs towards terminally differentiated neurons. JNK complexes specifically bind and activate the promoters of neurodevelopmental genes and inhibition of JNK leads to severely impaired neuronal differentiation, identifying JNK proteins as epigenetic regulators during neuronal maturation (Tiwari et al, 2011). We hypothesize that loss of JIP2 negatively impacts complex formation with JNK, which

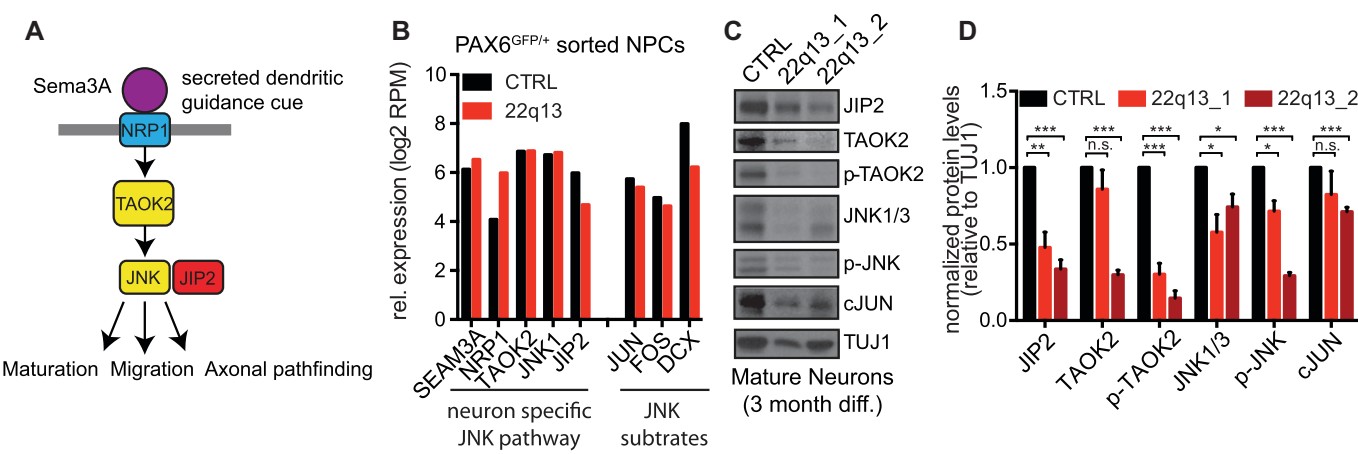

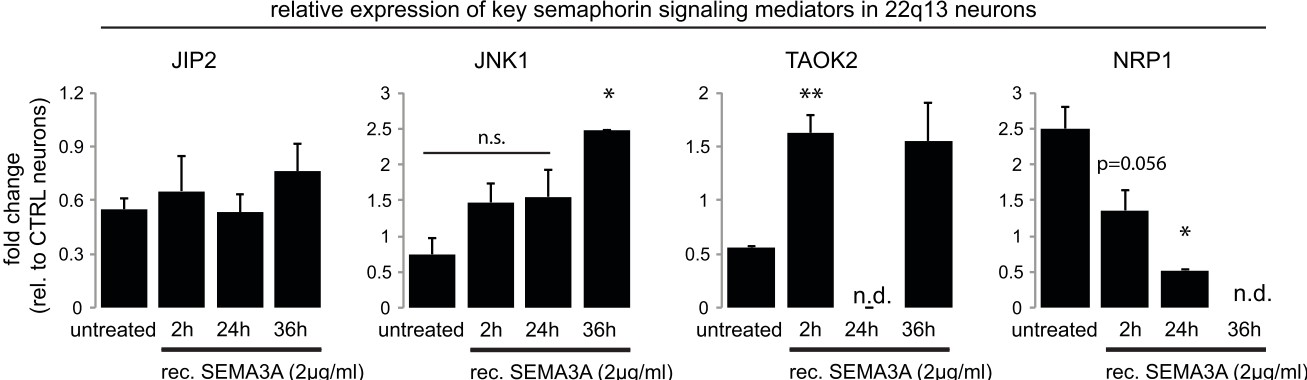

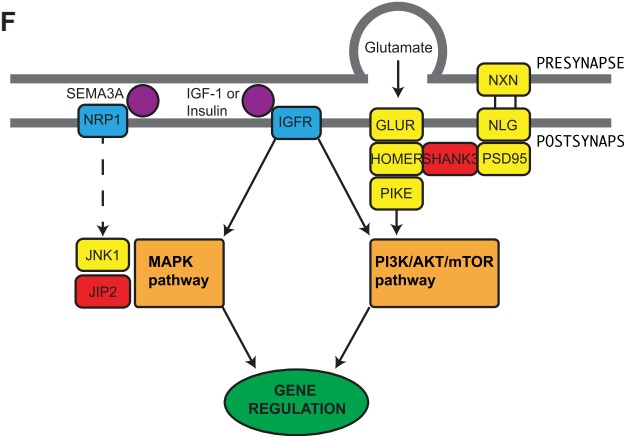

**Figure 4. Pharmacological activation of the NRP1 recues JNK expression in maturing patient neurons.**
**(A)** Mechanistic scheme of the semaphorin pathway in neurons. **(B)** Comparative transcript analysis (RNAseq) of key factors of semaphorin pathway and JNK target genes. **(C)** Protein analysis of mature neurons derived from control and two independent patient iPSC lines. **(D)** Quantification of immunoblots shown in (C). Data are represented as mean ± SEM (n = 3). **(E)** Transcript analysis (qRT–PCR) of key factors of the semaphorin pathway upon treatment with recombinant SEMA3A. Data are represented as mean ± SEM (two biological replicates including three technical replicates each). Two-tailed paired $t$ test, *$P$ < 0.05, **$P$ < 0.005. **(F)** Mechanistic molecular model for 22q13DS.

consequently contributes to a neurodevelopmental defect. Notably, NRP1, a semaphorin-specific receptor in developing neurons, was found to be up-regulated in patient NPCs and neurons. Interestingly, a genome-wide association study of a large cohort of ASD families identified SEMA5A as an ASD risk gene (Weiss and Arking et al, 2009).

The observed increase in NRP1 in our study might be a compensatory mechanism to maintain activity levels of the semaphorin pathway as needed for properly regulated neuronal maturation. Conversely, we also found that pharmacological stimulation of NRP1 restored JNK levels in neurons and NRP1 itself returned to

control levels. This suggests a regulatory feedback loop in which JNK impacts NRP expression and vice versa. We did not observe significant up-regulation of JIP2 upon NRP1 stimulation. Although it is possible that expression of the wild-type allele is increased, a more complex gene regulation may prevent specific induction of JIP2 in patient cells. However, genetically controlled knockout experiments in future studies are needed to ascertain this theory.

Fig 4F depicts a mechanistic model (modified and expanded from Kelleher et al [2012]) where loss of JIP2 impairs neuronal maturation by impacting the function of JNK and the MAPK pathway early in neurodevelopment. In addition, loss of SHANK3 results in impaired synaptic transmission and the alleviated activity of the AKT/mTOR pathway later in mature neurons. This model also takes into consideration that IGF-1 (and insulin) triggers the activation of the AKT/mTOR pathway and the MAPK pathway and is, therefore, able to ameliorate some 22qDS phenotypes. Earlier studies have shown that IGF-1 treatment improves cellular and behavioral phenotypes and even disease-specific symptoms in patients (Bozdagi et al, 2013; Shcheglovitov et al, 2013; Kolevzon et al, 2014).

Modeling complex neurological diseases such as 22qDS is challenging and requires human disease–relevant systems as animal models might not be optimal to understand specific and potentially subtle phenotypes of human disease. Further investigation is needed to carefully dissect the independent roles of SHANK3 and JIP2. Although previous studies focused on iPSC lines from patients that did not present with JIP2 deletions, it would be useful to engineer isogenic ESC lines that carry a deletion for either *SHANK3* or JIP2. Here, we began to study the involvement of JIP2 in 22qDS and although isogenic knockout ESCs provided interesting data, genetic rescue experiments, reestablishing JIP2 expression in those knockout lines, would help to consolidate our observation.

Additional genetically defined approaches would help to generate robust answers regarding the involvement of both candidate genes. In summary, our study provides suggestions that might further our current understanding of underlying mechanistic processes in 22qDS and suggests neuropilin receptors as potential additional therapeutic targets. However, future studies are required to strengthen our preliminary findings regarding the involvement of JIP2 in Phelan–McDermid syndrome.

# Materials and Methods

### IPSC induction and neural differentiation

22qDS patient fibroblasts were acquired from the RUCDR cell repository (http://www.rucdr.org/) and reprogrammed using the Stemgent RNA reprogramming kit (Warren et al, 2010). Briefly, patient fibroblasts were plated on human newborn foreskin fibroblast feeder cells in varying densities to determine the most efficient reprogramming condition and transfected daily with the reprogramming cocktail for about 2 wk (see Stemgent instructions). Clones were picked manually and expanded on inactivated mouse fibroblasts. WIBR3, a human ESC line described previously (Lengner et al, 2010), was used as control line. All cell lines used and generated are summarized in Table 1. For neural induction and cortical neuron differentiation, PSCs were exposed to dual SMAD inhibition combined

**Table 1. Overview and background information of all cell lines generated.**

| CTRL (hESC) | Gender | Age | | | | | No. of clones | Induction method | Diff. | PAX6-GFP |
|---|---|---|---|---|---|---|---|---|---|---|
| WIBR3 | Female | Blastocyst derived (Lengner et al, 2010) | | | | | 1 | N/A | Yes | Yes |

| Rutgers ID (iPS lines) | Gender | Age | Category | Location (deletion) | Ref. genome | Del. start | Del. end | Size (bp) | No. of clones | Induction method | Diff. | PAX6-GFP |
|---|---|---|---|---|---|---|---|---|---|---|---|---|
| MH0148130 (22q13_1) | Female (patient 1) | 35 | Terminal deletion | 22q13.3 | NCBI36, HG18 | 49210245 | 49691432 | 481187 | 4 | RNA | Yes | Yes |
| MH0148137 | Female | 12 | Terminal deletion | 22q13.3 | NCBI36, HG18 | 49479705 | 49522658 | 42953 | 4 | RNA | No | No |
| MH0148157 (22q13_2) | Male (patient 2) | 11 | Terminal deletion | 22q13.3 | NCBI36, HG18 | 49369190 | 49463084 | 93894 | 4 | RNA | Yes | No |

| Isogenic ESCs | Gender | Age | Category | Location (deletion) | CRIPSR Seq. | Clones picked | Assay | Size (del.) | No of clones | Induction method | Diff. | PAX6-GFP |
|---|---|---|---|---|---|---|---|---|---|---|---|---|
| WIBR3-1 (Δ22-1) | Female | N/A | Terminal deletion | 22q13.3 | Guide I: 5'-TAATGTCACTTTTGCTGCAGTGG-3' | 30 | PCR | ~10 kb | 1 | N/A | Yes | Yes |
| WIBR3-2 (Δ22-2) | Female | N/A | Terminal deletion | 22q13.3 | | 30 | PCR | ~10 kb | 1 | N/A | Yes | Yes |
| WIBR2-1 (Δ22-3) | Male | N/A | Terminal deletion | 22q13.3 | Guide II: 5'-CTGCTTGCCTGGGCTCCAGCTGG-3' | 30 | PCR | ~10 kb | 1 | N/A | No | No |

with established optimized protocols for cortical in vitro differentiation (Chambers et al, 2009; Shi et al, 2012a, b). PAX6-GFP reporter lines were differentiated for up to 4 wk and were subjected to FACS upon up-regulation of GFP. Mature cortical neurons were derived from PSC-derived NPC by extended differentiation in N2/B27 medium for up to 3 mo. Differentiation stages were identified by mor-phological characteristics and cell type–specific marker expression.

### Electrophysiological characterization of PSC-derived neurons

To analyze electrophysiological properties of maturing neurons, we used Axion Multi-Electrode-Arrays (https://www.axionbiosystems.com/). Electrode-containing wells were coated with poly(ethyleneimine) solution (P3143; Sigma-Aldrich) and Matrigel (#47743; VWR). Approximately 120,000 immature neurons were plated directly on top of the electrode arrays, and cells were allowed to settle for 20 min, after which cells were cultured in 1 ml N2B27 medium for up to 3 mo. The medium was changed every 2 d. Dependent on the intensity of detected field potentials, arrays were measured two to four times a week. Data were recorded for 300 s per session. Data were acquired using AxIS software and analyzed using NeuroMetrix and Prism GraphPad.

### Genome editing and Southern blot

CRISPR/Cas9 plasmids were cloned as described earlier (Cong et al., 2013; Mali et al., 2013) (Addgene). The PAX6-GFP donor plasmid was cloned by using a p2A-PKG-Puro backbone as described in Hock-emeyer et al (2009). PSCs were electroporated with both the CRIPSR plasmid (10 μg) and the donor vector (25 μg), and puromycin selection was applied for 14 d. Surviving clones were manually picked and subjected to PCR and Southern blot analysis. For the latter, external and internal probes were designed and radioactively labeled and enzymatically digested genomic DNA was blotted and probed against. Correctly targeted clones were selected, propagated, and differentiated.

### Immunoassays

Standard immunocytochemistry and Western blot techniques were used to characterize cell type–specific markers and phenotypically relevant protein levels. The following antibodies were used for immunostaining according to the manufacturers' instructions: anti-GFP (#1010; Aves Labs Inc), anti-PAX6 (PRB-278P; Covance), anti-Nestin (PRB-315C; Covance), anti-Map2 (#M2320; Sigma-Aldrich), and anti-Ctip2 (ab18465; Abcam). The following antibodies were used for Western blot assays: anti-Shank3 (sc30193; Santa Cruz), anti-Shank3 (sc 23547; Santa Cruz), anti-PSD95 (ab2723; Abcam), anti-Homer (#160011; Synaptic Systems), anti-NLGN1 (SAB1407153; Sigma-Aldrich), anti-PIKE (#07-675; Upstate), anti-Akt (#4691S; Cell Signaling), anti-phospho Akt (#4058; Cell Signaling), anti-mTOR (#4691; Cell Signaling), anti-phospho mTOR (#2971; Cell Signaling), anti-Jip2 (N135/37; Neuro-Mab), anti-NRP1 (SAB1411572; Sigma-Aldrich), anti-JNK1/3 (sc474; Santa Cruz), anti-phosphor JNK (#V793A; Promega), anti-DCX (#4604; Cell Signaling), anti-NeuN (#MAB377; EMD Millipore), anti-Tuj1 (#903401; BioLegend), anti-GAPDH (sc 47724; Santa Cruz), and anti-Actin (sc1616; Santa Cruz). Western blot quantification was performed using ImageJ.

### Quantitative RT–PCR (qRT-PCR) and RNAseq

RNA was isolated using Trizol (#15596026; Thermo Fisher Scientific) or Total RNA Kit (R-6834-01; Omega). For qRT–PCR, cDNA was synthesized using SuperScript III First Strand Synthesis Kit (#18080051; Thermo Fisher Scientific).

RNA for RNAseq runs was validated by Agilent Bioanalyzer. RNAseq was performed on a Illumina HiSeq (SMARTer Ultra-low POLY A, single end reads). Data analysis was performed using SeqMonk (http://www.bioinformatics.babraham.ac.uk/projects/seqmonk) and R (https://www.r-project.org/).

### Accession codes

Raw data from RNAseq analyses have been deposited in the NCBI Gene Expression Omnibus under the accession number GSE90726.

## Supplementary Information

## Acknowledgements

We thank R Alagappan, D Fu, and T Lungjangwa for their technical support with human PSC cultures. We would like to thank P Wisniewski, C Zollo, C Arano, and M Ly of the Whitehead Institute FACS-core facility. J Love and S Gupta of the Whitehead Genome Technology Core helped with RNAseq. We would like to thank all members of the Jaenisch laboratory for helpful discussions and comments on the manuscript. This work was supported by a grant from the Simons Foundation to the Simons Center for the Social Brain at Massachusetts Institute of Technology. R Jaenisch was supported by National Institute of Health grants 1R01NS088538-01, HD 045022, and 2R01MH104610-15.

## Author Contributions

R Roessler: conceptualization, data curation, investigation, methodology, writing—original draft, review, and editing.
J Goldmann: methodology.
C Shivalila: methodology.
R Jaenisch: conceptualization and writing—review and editing.

### Conflict of Interest Statement

The authors declare that they have no conflict of interest.

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
